# Myofibrillar Protein Interacting with Trehalose Elevated the Quality of Frozen Meat

**DOI:** 10.3390/foods11071041

**Published:** 2022-04-04

**Authors:** Shijie Xu, Ping Li, Fei Han, Hui Zhou, Kai Zhou, Ying Wang, Kezhou Cai, Cong Li, Baocai Xu

**Affiliations:** 1School of Food and Biological Engineering, Hefei University of Technology, Hefei 230601, China; xu_shijie_1997@163.com (S.X.); pingli@mail.hfut.edu.cn (P.L.); hf19970309@163.com (F.H.); kaizhou@hfut.edu.cn (K.Z.); wywendy8899@126.com (Y.W.); kzcai@hfut.edu.cn (K.C.); cobralc@126.com (C.L.); baocaixu@163.com (B.X.); 2State Key Laboratory of Meat Processing and Quality Control, Jiangsu Yurun Meat Food Co., Ltd., Nanjing 210000, China; 3Engineering Research Center of Bio-Process, Ministry of Education, Hefei University of Technology, Hefei 230601, China

**Keywords:** myofibrillar protein, trehalose, chitooligosaccharide, interaction, meat quality

## Abstract

This work studied the interactions between trehalose/chitooligosaccharide (COS) and myofibrillar protein (MP), and the effect of such interactions on the quality of meat after freezing was also evaluated. Fourier transform infrared spectroscopy showed that both trehalose and COS could enhance the content of hydrogen bonds of MP. Zeta potential measurement displayed trehalose/COS reduced the absolute value of the surface potential of MP. The results of Raman spectroscopy suggested that the hydrophobic residues of MP were more exposed after treatment with trehalose/COS. Thus, trehalose and COS could both interact with MP through non-covalent bonds. Subsequently, the evaluation of the effect of trehalose and COS on the physicochemical properties of frozen meat was conducted. Results showed that both trehalose and COS significantly reduced thawing loss of frozen meat, and sensory evaluation showed that trehalose had a better performance from the perspective of smell, texture, and overall consumer acceptance. In conclusion, trehalose/COS interacting with MP can reduce meat thawing loss, which might provide technical guidance in the quality control of frozen meat.

## 1. Introduction

Freezing is a commonly used storage method for meat because it can inhibit the proliferation of spoilage microorganisms [1,2]. However, the formation of large ice crystals could take place during freezing, which would seriously damage muscle fibers and cause a considerable amount of juice loss [3,4]. Myofibrillar protein (MP) plays a critical role in meat, for its ability to form cohesive structures and firm texture [5,6]. However, during frozen storage, serious MP denaturation occurs [7,8]. Such denaturation of MP would result in a serious amount of juice loss [4]. A number of methods and technologies have been developed to reduce the negative effect of frozen storage on meat quality. For example, a high voltage electrostatic field can be utilized for meat thawing, by which a significant improvement including higher freezing efficiency and lower thawing loss can be achieved [9]. However, safety concerns and relatively high costs limit its broader application. The effective and low-cost solution for frozen meat quality improvement remains to be explored.

Trehalose is a non-reducing disaccharide consisting of two glucose molecules condensed through the hemiacetal hydroxyl group, connected through a 1,1-glycosidic bond [10]. Trehalose is an effective protein and cell membrane stabilizer, able to enhance cell’s resistance towards stress environments such as overpressure, high temperature, freezing, dehydration, and hypertonicity [11,12,13,14]. When cells are dehydrated, the hydroxyl group of trehalose can replace water molecules to form hydrogen bonds with the protein on the cell membrane. During the dehydration, the lipid membrane experiences the transition from the liquid crystal phase to the gel phase. During the aprotic process, the reverse transition process causes leakage of the lipid membrane. Trehalose can reduce the phase transition temperature of the membrane, effectively preventing leakage [15,16,17,18]. Trehalose has been applied to plenty of food products such as shrimp muscle, surimi, dough as cryoprotectants to reduce juice loss, maintain firm texture, and stabilize protein structures [19]. Chitooligosaccharide (COS) is an alkaline amino oligosaccharide consisting of N-acetylglucosamine connected by β-1,4 glycosidic bonds [20]. Similar to trehalose, COS also showed a well-improved effect on cell resistance toward stress conditions such as chilling and salt [21,22]. In addition to the cell-protective effect mentioned above, COS showed a protective effect on biomacromolecules such as proteins under stress conditions as well [20]. While trehalose and COS are emerging stabilizers for cell protection during freezing, their potential in improving frozen meat quality has rarely been explored.

Hydrogen bond, electrostatic interaction, and hydrophobic interaction are the main interactions between protein and saccharide molecules [23]. Given the critical role MP plays in forming cohesive structures and retaining water in meat [5,6], the study of these interactions would help to explore the potential of trehalose and COS as a cryoprotectant for frozen meat. Herein, this work firstly investigated the interactions between trehalose/COS and MP. The effect of such interactions on meat quality, including thawing loss and sensory characteristics, was then evaluated. The results might provide guidance for reducing the quality loss of meat after freezing.

## 2. Materials and Methods

### 2.1. Sampling

Fresh pork from *biceps femoris* post-mortem within 48 h was purchased from local Walmart (Walmart, Hefei, China), and transported to the laboratory within 30 min at 0~4 °C. The surface fascia, connective tissue, and fat were trimmed away. The pork was cut into rectangular samples of 15 cm × 5 cm × 2 cm, weighing 150 g ± 7.5 g.

Trehalose and chitooligosaccharide (COS) samples were purchased from Qingdao Bozhi Huili Biological Co. Ltd. (Qingdao, China). Samples were food-grade, 98% purity, 1000 mesh size. All chemical reagents were analytically pure.

### 2.2. Extraction of Myofibrillar Proteins (MPs)

MPs were extracted from fresh pork (*biceps femoris*) isolates according to a previously described method [24]. Briefly, the trimmed meat was minced and then homogenized three times with four volumes (*w*/*v*) of isolate buffer (0.1 M NaCl, 2 mM MgCl_2,_ 1 mM EGTA, and 10 mM sodium phosphate, pH 7.0). The homogenized liquid was centrifuged at 2000× *g* for 15 min to obtain a crude MP pellet, which was washed three times with 10 mM sodium phosphate buffer (pH 7.0) containing 0.1 M NaCl. In the last wash, the liquid was filtered through four layers of cheesecloth to remove connective tissue and visible fat. After the pH of the myofibril suspension was adjusted to 6.5 with 0.1 M HCl, the suspension was centrifuged. Finally, MP isolations were obtained, kept on ice, and used within 48 h. The concentration of MP was determined by the biuret method using bovine serum albumin.

### 2.3. MP Interacting with Trehalose/COS

The MP was diluted to 10 mg/mL with PIPES buffer (pH 7.0), into which an equal volume of 10 mg/mL trehalose or COS was added under mechanical agitation. The mixture solution was then incubated under magnetic stirring at 0~4 °C for 12 h, freeze-drying under 50 mPa at −50 °C for 24 h, the powder samples containing MP only, MP and trehalose, and MP and COS were finally obtained and tagged as MP, MP + trehalose, and MP + COS, respectively.

### 2.4. Fourier Transform Infrared (FTIR) Spectrum

FTIR analysis was carried out as reported by Velazqueza [25] and Yang et al. [26]. with some modifications. At room temperature (20 ± 2 °C), the MP, MP + trehalose, and MP + COS were mixed with KBr, ground, and tableted. FTIR spectra of samples (1 mg) were recorded using an infrared spectrometer (Spectrum 400. PerkinElmer Inc., Waltham, MA, USA) equipped with an ATR prism crystal accessory. Data were recorded at 650~4000 cm^−1^ applying 16 scans at 4/cm^−1^. Spectrum software, version 6.3.2 (PerkinElmer Inc., Waltham, MA, USA), was used to remove and filter background interference. At least three replicates were carried out for each sample in the FTIR scan.

### 2.5. Zeta Potential

The Zeta potential of MP, MP + trehalose, and MP + COS was determined as reported by Wu et al. [27] with some modifications. The samples were diluted to 1 mg/mL with PIPES buffer (pH 7.0), the solution samples were then injected into the Zeta potentiometer (Zatasizer Nano-ZS90, Malvern, UK). Test parameters included: scattering angle, 90°; equilibrium time, 60 s; test temperature, 25 °C. At least three replicates were carried out for each sample.

### 2.6. Raman Spectroscopy

Raman experiments were carried out as previously reported by Monago-Maraña [28] with some modifications with a HORIBA JOBIN YVON LabRam HR Evolution (Horiba/Jobin. Yvon, Longjumeau, France) at room temperature (20 ± 2 °C). The excitation laser beam (532 nm exciting line of a Spectra Physics Ar-laser) was focused on the samples through an optical microscope equipped with a 50× lens. The laser power applied to the sample surface was adjusted to 5 mW to avoid possible damage to the samples. During the experimental scanning, samples were smeared on a glass slide. The spectra were recorded at the range of 500~2000 cm^−1^. The spectrum was obtained through the following methods: 5 scans, 120 s exposure time, 1 cm^−1^ resolution. The scanning speed was 120 cm^−1^/min with data recorded every 1 cm^−1^. Acquisition of 1 spectrum continued for about 2 min. At least three replicates were carried out for each sample in the Raman scan. The Spectra obtained as described above were smoothed and baseline corrected using Labspec5 (Horiba/Jobin. Yvon, Longjumeau, France).

### 2.7. Treatment of Meat Samples

After shaping and weighing, meat pieces were randomly assigned to three different types of treatment as follows: add trehalose (0.5%, *w*/*w*), add COS (0.5%, *w*/*w*), and blank control (no adding). The treating process was then carried out. The trehalose/COS was evenly sprinkled on the surface of the cuboid meat pieces. Subsequently, the meat pieces were wrapped with polyethylene (PE) film (O_2_ transmittance 600 cm^3^·m^−2^·d^−1^, moisture transmittance 5~10 g·m^−2^·d^−1^). The blank control samples were also wrapped with the same PE film. The blank control, meat treated with trehalose, and meat treated with COS were tagged as meat, meat + trehalose, and meat + COS, respectively. Three replicates of each sample were made for each type of treatment.

All the samples were frozen in a refrigerator (Haier, Qingdao, China) at −30 °C for 24 h, then taken out of the refrigerator and thawed at 0~4 °C for 12 h, and the core temperature of the meat was raised to 0~4 °C.

### 2.8. Thawing Loss

The meat was weighed and recorded as m_0_ after cutting and shaping. The weight after adding trehalose/COS was recorded as m_1_. After freezing and thawing, the meat was weighed again and recorded as m_2_. The thawing loss was calculated as [(m_0_+m_1_) − m_2_]/[m_0_+m_1_]. m_1_ was 0 for the untreated meat without trehalose or COS. Three replicates of each sample were made for each type of treatment. The experiment was carried out three times.

### 2.9. Ice Crystals in Meat Muscle

The ice crystal morphology was observed by light microscopy following method reported by Xie, et al. [29] and Kaale and Eikevik [30]. Frozen meat samples (10 mm × 10 mm × 10 mm), which had been cut perpendicularly to the muscle fibers, were fixed with 3% glutaraldehyde in sodium phosphate buffer (0.1 M, pH 7.4) for 24 h. The samples were then dehydrated at 4 °C with ethanol solutions of gradient concentrations (70–100%, *v*/*v*) using a dehydrator (JJ-12 J, Junjie, China). Then, the tissues were immersed in xylene for transparency. Wax was used to soak the sample to ensure the fixation of the meat tissue. The samples were then embedded in paraffin in an embedding station (JB-P5, Junjie, China) to facilitate slicing. Slices were obtained using a Pathology slicer (RM2016, Leica, Germany) and were subjected to hematoxylin and eosin (HE; G1005, Servicebio), followed by observation at 200× magnification with a Light microscope (E100, Nikon, Japan).

### 2.10. Surface Color

The surface color of the meat was measured according to the CIE *L***a***b** color system (*L** for brightness, *a** for redness, and *b** for yellowness) using a colorimeter (WSF colorimeter, INESA Optical, Shanghai, China) equipped with a d/65 light source. Three replicates of each sample were made for each type of treatment. The experiment was carried out three times.

### 2.11. pH

The pH value of the meat was measured using a Testo205 pH meter (Testo AG, Titisee-Neustadt, Germany) equipped with a glass probe (automatic adjustment for temperature). The measurements were carried out five times for every meat piece, the positions in which the probe was inserted were the four corners and the center of the meat’s upper surface. Three replicates of each sample were made for each type of treatment. The experiment was carried out three times.

### 2.12. Sensory Evaluation

Meat samples were assessed by a sensory evaluation panel. The evaluating experiment was carried out in individual booths at the Sensory Analysis Laboratory, with the participation of 60 untrained judges of both male and female gender, belonging to the Meat Process & Health Innovation Team of the Hefei University of Technology. The meat samples were thawed and then immediately presented to the panelists in random order. Each judge received an evaluation score form along with triplicate samples from all types of treatment, placed in a stainless steel plate. Judges were instructed to evaluate the color, smell, texture, and overall acceptance on a nine-point scale. The definition and score criterion of sensory quality attributes is presented in Table 1 [31,32]. The experiment was carried out three times.

### 2.13. Statistical Analysis

Statistical analysis of results was performed using SPSS software (SPSS Inc., Chicago, IL, USA, Ver.26). All data were presented as means ± standard deviation (SD). One-way analysis of variance (ANOVA) was employed to determine the statistical difference. Significant differences between means were identified using Duncan’s multiple range test (*p* < 0.05). At least three replicates of each sample were made for each type of treatment in the experiment. The experiment was carried out three times.

## 3. Results and Discussion

### 3.1. The Interaction of Myofibrillar Protein (MP) with Trehalose and COS

#### 3.1.1. FTIR

The Fourier-transformed infrared spectra (FTIR) of the trehalose, COS, MP, MP + trehalose, and MP + COS are shown in Figure 1. For trehalose, the peak at 3271 cm^−1^ is characteristic of the stretching vibration of -OH, the peak at 2928 cm^−1^ is the stretching vibration of C-H, the peak at 1452 cm^−1^ is the bending vibration of -CH, the peak at 1240 cm^−1^ is the stretching vibration of C-O, and the peak at 995 cm^−1^ is C-H connected to glycosidic bonds [33]. For COS, the characteristic peaks mainly appeared at 3255 cm^−1^, 2885 cm^−1^, 1623 cm^−1^, 1520 cm^−1^, 1418 cm^−1^, and 1066 cm^−1^, which were similar to trehalose. The N-H stretching vibration absorption at 3500~3100 cm^−1^, the stretching vibration absorption of C=O from -COO- at 1623 cm^−1^, and the C-N stretching vibration absorption at 1066 cm^−1^ appeared to be the main difference between trehalose and COS.

For MP, its characteristic absorption peaks appeared at 3285 cm^−1^, 2925 cm^−1^, 1635 cm^−1^, 1535 cm^−1^, 1401 cm^−1^, and 1073 cm^−1^, corresponding to amide III O-H stretching vibration, C-H stretching vibration, carbonyl C=O double bond stretching vibration, C-N stretching, N-H bending vibration, and C-O stretching vibration, respectively [34].

For MP + trehalose and MP + COS, the FTIR spectrum of trehalose and COS has been removed by differential spectroscopy [35,36]. The characteristic peaks mainly appeared at 3280 cm^−1^, 2930 cm^−1^, 1645 cm^−1^, 1545 cm^−1^, 1400 cm^−1^, and 1076 cm^−1^. The characteristic absorption peaks of the FTIR spectrum of MP + COS mainly appeared at 3285 cm^−1^, 2930 cm^−1^, 1641 cm^−1^, 1540 cm^−1^, 1400 cm^−1^, and 1070 cm^−1^ [35,36]. No appearance of novel peaks or disappearance of original peaks of MP was observed after it interacted with trehalose or COS. However, the main characteristic peaks of the MP shift might suggest a strong electrostatic interaction between trehalose/COS and MP. After interacting, the FTIR absorption of amide II, amide I, and -OH at 1500–1600 cm^−1^, 1600–1700 cm^−1^, 3000–3700 cm^−1^ increased significantly. This might be attributed to the hydrogen bonds between trehalose/COS and MP. Therefore, the content of intramolecular and intermolecular hydrogen bonds of MP increased and caused the increase of the content of the above structure [26,27,37].

MP contains a large number of hydrogen bond donors and receptors, such as nitrogen atoms, ether oxygen, and the phenyl ring of phenylalanine, which may interact with trehalose/COS hydroxyl groups through hydrogen bond [35,36]. Gaussian fitting was performed at the range of 3700~3000 cm^−1^. The results are shown in Figure 2. The attribution, distribution, and intensity of the sub-peak of the hydrogen bond were obtained. The absorption peaks at 3595 cm^−1^ correspond to free hydroxyl groups. The absorption peaks at 3409 cm^−1^ and 3219 cm^−1^ correspond to intramolecular hydrogen bonds. The absorption peaks at 3522 cm^−1^, 3295 cm^−1^, and 3132 cm^−1^ correspond to intermolecular hydrogen bonds. The area of each sub-peak was calculated to obtain the relative content of various hydrogen bonds, as was shown in Table 2.

As was presented in Table 2, the main component of the intramolecular hydrogen bond of MP was the hydroxyl-hydroxyl interaction, while the main component of the intermolecular hydrogen bond was the hydrogen bond between hydroxyl and ether oxygen [38,39]. After interacting with trehalose/COS, the component of the MP hydrogen bond tended to change in similar trends as the content of free hydroxyl groups, intermolecular hydrogen bonds including the fact that OH...π and OH...N decreased, and the content of intermolecular hydrogen bond OH...ether O and intramolecular hydrogen bond annular polymer increased. This trend may be attributed to the introduction of a large number of pyranose structures from trehalose/COS [35,36]. Hydrogen bonds were the main interactions that maintain the spatial structure of MP. Trehalose and COS had a great impact on the intensity and distribution of hydrogen bonds of MP. Trehalose and COS may bind to active sites such as -OH and benzene ring on MP through hydrogen bond, which will affect their water retention ability after frozen storage [40].

#### 3.1.2. Zeta Potential

Zeta potential is the potential of the surface shear layer of charged particles that can be utilized in the description of electrostatic interaction between colloidal particles. The effect of trehalose/COS on the potential of MP is shown in Figure 3. Interacting with trehalose/COS had a considerable effect on the potential of MPs. The absolute value of MP + trehalose and MP + COS were significantly lower than the natural MP (*p* < 0.05). The zeta potential of MP + trehalose was lower than MP by approximately 7.36. When the COS was introduced into the complex, its surface potential had transformed into positive, and its positive charge value eventually reached 15.43, which was carrying a positive charge not only more than MP but also more than COS. This gap supposedly resulted from the difference between the nature of trehalose and COS. Trehalose consists of two glucose molecules [10], which do not carry a charged group, thus, it conducts weaker electrostatic interaction with MP. While COS carried -NH_2_ [20], which would form a much stronger electrostatic interaction with negatively charged MP. It was suggested that the -NH_2_ of COS bound to the active position of MP through electrostatic interaction, and other positively charged groups carried by COS were concentrated on the surface of colloids, hence causing further climbing of the positive charge quantity of MP + COS.

Almost all charged groups in protein molecules are distributed on their surface. In the process of protein aggregation, electrostatic interaction is usually manifested as mutual repulsion, and the zeta potential value was negative, indicating that MP was negatively charged. The introduction of trehalose/COS reduced the surface zeta potential of MP, thereby reducing the electrostatic repulsion of MP. It may improve the hydrogen bond state within the MP [23,27].

#### 3.1.3. Raman Spectroscopy

Raman spectroscopy scan was conducted to investigate the micro environment surrounding amino acid side chains of MP. The intensity of the phenylalanine ring at 1003 cm^−1^ respiratory vibration does not change with the protein structure, hence it can be used for internal standards to investigate the tryptophan stretching vibration at 760 cm^−1^, which reflected the changes in the hydrophobic interaction between protein molecules [41]. The ratio of normalized strength I760/I1003 of MP increased after it interacted with trehalose/COS, indicating that the hydrophobic interaction between MP was enhanced. As shown in Figure 4, The interaction between MP was strengthened, and the interaction between MP and water was weakened. The enhancement of hydrophobic interaction between MP molecules after the introduction of trehalose and COS suggested that trehalose/COS may be able to weaken the interaction between protein molecules and water molecules by enhancing the interaction within protein molecules [37,42].

### 3.2. Effect of Trehalose and COS on Meat Quality

#### 3.2.1. Thawing Loss

The effect of adding trehalose/COS to frozen meat thawing loss was shown in Figure 5. In this work, the thawing loss was about 3.19%. After treatment with trehalose or COS, the thawing loss of meat decreased by 42.63% and 51.41%, respectively. The juice loss of meat caused by freezing reduced considerably. During the freezing process, the formation of ice crystals caused serious mechanical damage to the muscle. Therefore, the water channels through which water migrates from the inside of the muscle fiber to its surface were enlarged [5,6]. The water-holding capacity of meat was highly related to the integrity of MP [4]. A large number of active groups such as hydroxyl groups are possessed by trehalose and COS, which could connect to MP through hydrogen bonds and electrostatic interactions as shown above. The influence of trehalose/COS on the MP helped to protect its structure, and eventually reduced the thawing loss of frozen meat [11,12,13,43].

#### 3.2.2. Ice Crystals in Meat Muscle

After the meat was frozen, the formation of ice crystals caused mechanical damage, resulting in a decrease in the water-holding capacity of the meat. To this end, the effect of trehalose/COS on ice crystals was observed. Figure 6 shows the hematoxylin-eosin (HE) staining section of meat. As shown in Figure 6, the muscle fibers of the fresh meat were regular, tightly arranged, and intact, and the gaps between the muscle fibers were small. After the freezing, the muscle fibers were twisted and deformed. The gaps which were considered to be the locations of ice crystals between the fibers were enlarged. In this case, the water intercepted between the muscle fiber flowed out, generating juice loss during the thawing process. After adding trehalose/COS, the crystal size was smaller, and the structure of muscle fibers tended to be intact. The protective effect of trehalose/COS on crystals size may be the reason why it can reduce meat thawing loss [34,39,40,44].

#### 3.2.3. Surface Color

Color has a significant impact on consumers’ decision-making intention for meat purchase [45]. Changes in the surface color of meat from treated groups were shown in Table 3. As shown in Table 3, trehalose and COS both significantly increased the *L** value (*p* < 0.05). The *L** value of meat increased from the original 51.09 to 57.54 after adding trehalose and 57.48 after adding COS, respectively. This may be due to the fact that trehalose/COS made the juice ooze out of the meat surface, which enhanced the reflection of light [46]. Trehalose did not significantly affect the *a** value of meat, while COS caused a significant decrease in the *a** of the meat (*p* < 0.05). This may be attributed to the color of the trehalose solution (transparent) and the COS solution (dark brown) [47]. Both trehalose and COS increased the *b** value, the reason for which would be similar to that for the changes in the *a** value of the meat surface. Trehalose/COS presented good hydrophilicity [48]. After they were added to the surface of the meat, they were easily dissolved in the original juice on the surface of the meat [49].

#### 3.2.4. pH

The effect of trehalose and COS on the pH value of meat was shown in Table 3. Both trehalose and COS caused a significant drop in the pH value of meat samples (*p* < 0.05). After adding trehalose/COS, the pH value of meat dropped from 5.93 to 5.88 and 5.85, respectively. The pH value of the aqueous solutions of trehalose/COS were 5.78 and 5.57, respectively. It may be that the trehalose and COS made the surface of the meat form a higher concentration acidic solution, causing the pH value of the meat sample to drop [50].

#### 3.2.5. Sensory Evaluation

The results of sensory evaluation for frozen then thawing meat samples are shown in Table 4. The effect of trehalose and COS on the purchase intention of the consumer to buy meat were evaluated. Both trehalose and COS elevated the color scores of meat, which might be attributed to the enhanced reflection of light, making the meat more desirable. For the smell, trehalose increased the smell scores while the COS decreased the smell scores significantly (*p* < 0.05). Trehalose did not generate odor, thus the increasing scores might be due to the positive effect of trehalose on meat freshness. The byproducts when the COS was extracted from crustacean materials presented a fishy smell, which might result in an unpleasant smell. The texture of meat samples was improved after treatment with trehalose, while the COS significantly decreased the texture scores of meat samples (*p* < 0.05). This might be due to the different nature of water solutions on the meat surface after trehalose/COS was added. Overall, trehalose increased the sensory scores of meat, being proper for application in fresh meat. The negative effect of COS on meat quality suggested that finer extract technology and further study of its effect on meat quality were necessary [51,52].

### 3.3. Mechanism of Trehalose and COS Reducing Frozen Meat Thawing Loss

Considerable amount and intensity of interaction were seen between trehalose/COS and MP through hydrogen bonds and electrostatic interaction. Based on these, the mechanism of trehalose and COS reducing frozen meat thawing loss was proposed. Figure 7 shows the effect of trehalose and COS on meat, MP, muscle fiber, and ice crystals. In Figure 7a, the muscle fibers in their natural state without the addition of trehalose or COS were arranged tightly and orderly well in parallel to each other. The water within the muscle was separated and retained by the membrane structure, maintaining a stable state. Figure 7d presented the state of meat treating with trehalose or COS. As shown in Figure 7d, trehalose/COS are both bound to MP through hydrogen bond and electrostatic interaction, which appeared macroscopically as trehalose/COS attached to the active binding sites of MP on the surface and internal of muscle fibers. By binding to MP, the trehalose/COS partially replaced the water molecules originally bonded to MP, which would reduce the negative effect of water on MP during freezing [15,16,17,18,22]. The main difference between the effect of trehalose and COS was that COS contains a number of -NH_2_ groups carrying positive charge, which made the intensity of their interactions with negatively charged MP higher. Figure 7b shows the state of the meat not treated with trehalose/COS after being frozen. As shown in Figure 7b, a large number of ice crystals of bigger size were formed in frozen meat, and it seriously damaged muscle fibers. Figure 7e shows the frozen muscle that had been treated with trehalose/COS previously. Due to the effect of trehalose/COS on ice crystal growth, the ice crystals were smaller in size and more regular in shape, and the damage to the muscle was also less. Figure 7c,f showed the thawing process of frozen meat and meat treated with trehalose/COS, respectively. A large amount of juice loss occurred in frozen meat during thawing for the formation of large ice crystals and the muscle fiber deformation, while the meat treated with trehalose/COS had smaller water channels and less water loss due to the protective effect of trehalose/COS on muscle fiber structure and the ice crystals in smaller size [10,13,19,44].

## 4. Conclusions

Trehalose and COS could both interact with MP through hydrogen bonds and electrostatic interaction, which exposed the hydrophobic residues of MP. Such interactions could prevent the migration of water molecules during meat freezing, thereby promoting the formation of smaller ice crystals, and thus reducing the thawing loss of frozen meat after thawing. After treatment with trehalose or COS, the thawing loss of frozen meat was reduced by 42.63% and 51.41%, respectively. At the same time, trehalose and COS both reduced the pH value and enhanced the brightness of meat. From the view of sensory quality, meat treated with trehalose was comparable to fresh meat in the perspective of color, smell, and texture, which was significantly better than those treated with COS. Thus the quality of frozen meat was elevated after being treated with trehalose. Overall, both trehalose and COS could interact with MP to reduce the thawing loss of meat. However, trehalose was a more promising cryoprotectant for fresh meat considering the sensory quality.

## Figures and Tables

**Figure 1 foods-11-01041-f001:**
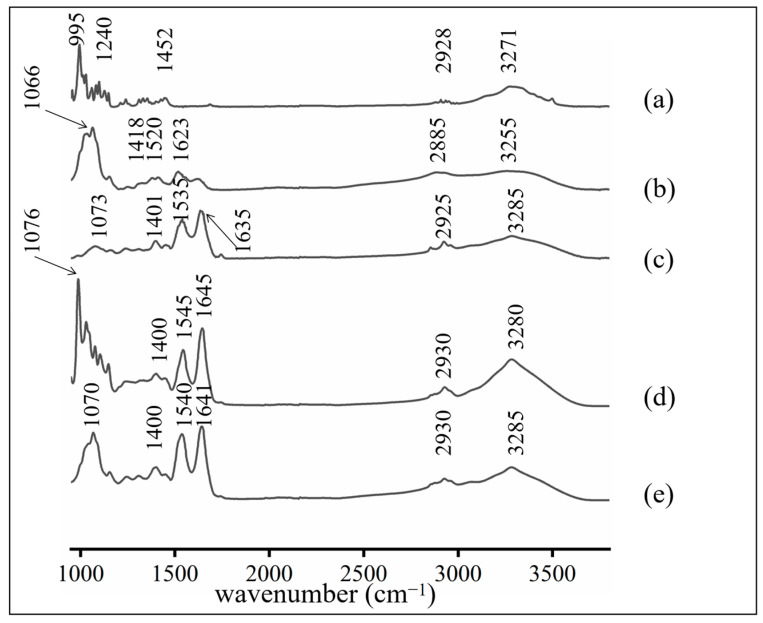
FTIR spectra of (**a**) trehalose, (**b**) COS, (**c**) MP, (**d**) MP + trehalose, and (**e**) MP + COS.

**Figure 2 foods-11-01041-f002:**
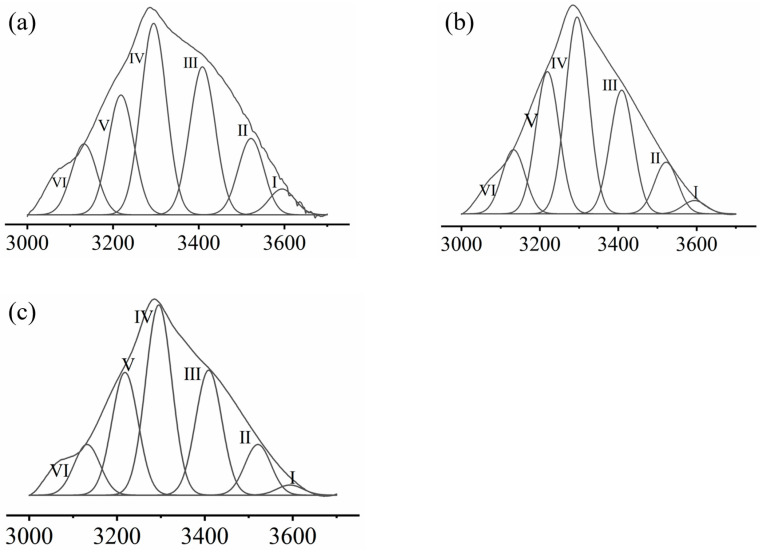
Gauss curve fitting of FTIR spectra of different samples, (**a**) MP, (**b**) MP + trehalose, and (**c**) MP + COS by different types of hydrogen bonds. Roman number I–VI is the serial number of sub-peaks corresponding to the hydrogen bond types in Table 2.

**Figure 3 foods-11-01041-f003:**
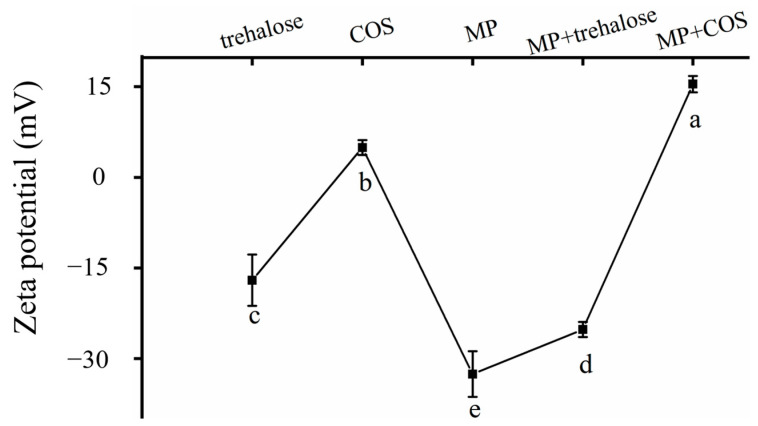
Zeta potential of trehalose, COS, MP, MP + trehalose, and MP + COS.

**Figure 4 foods-11-01041-f004:**
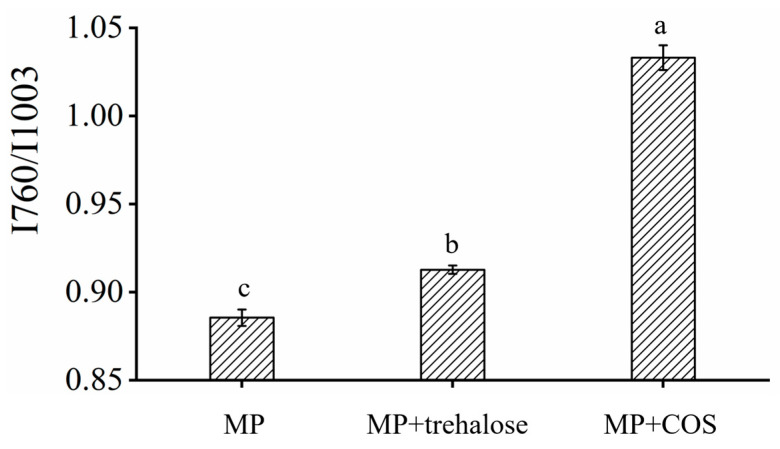
The normalized intensity of the I760/I1003 doublet bands of MP, MP + trehalose, and MP + COS. The letters (a–c) on the column indicate significant differences (*p* < 0.05) between the data of different samples.

**Figure 5 foods-11-01041-f005:**
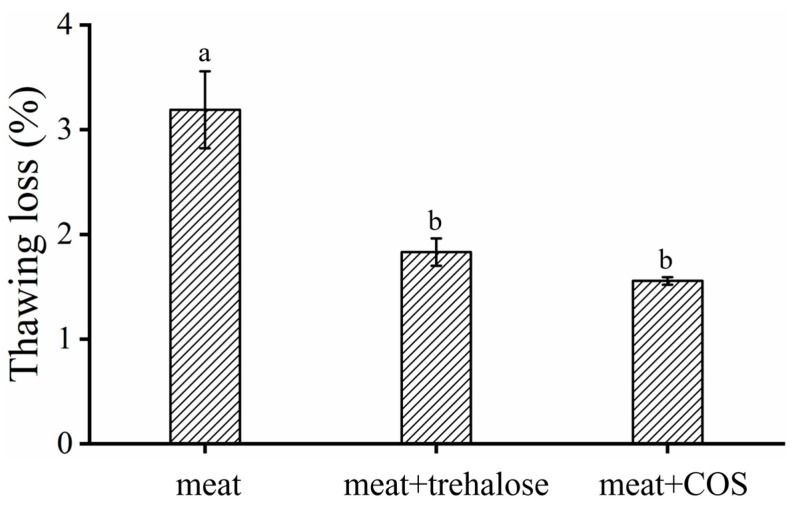
The effect of trehalose/COS on thawing loss of frozen meat. The results are mean ± standard deviation, and the letters (a,b) in the same column indicate significant differences (*p* < 0.05) between the data of different samples.

**Figure 6 foods-11-01041-f006:**
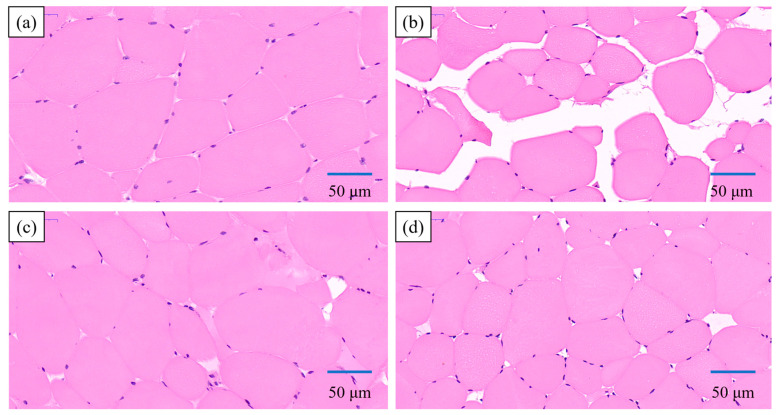
Observation of morphology and distributions of ice crystals in (**a**) fresh meat, (**b**) frozen meat, (**c**) frozen meat pretreated with trehalose (meat + trehalose), and (**d**) frozen meat pretreated with COS (meat + COS) samples after freezing in the view of the cross-section to the muscle fibers. The meat fibers were colored in red by eosin, and the nuclei of muscle cells were stained blue by hematoxylin, while the voids (white zones) represented the location of the ice crystals.

**Figure 7 foods-11-01041-f007:**
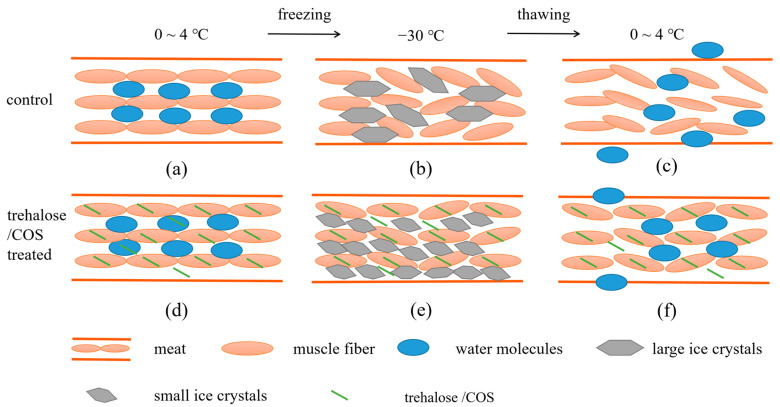
The schematic diagram of the underlying mechanisms of the trehalose/COS treatments affecting the thawing loss of frozen meat. In the image, (**a**–**c**) represented meat, (**d**–**f**) represented meat treated with trehalose/COS. (**a**,**d**) represented meat before frozen. (**b**,**e**) represented meat under frozen situation; (**c**,**f**) represented meat during thawing. The ice crystals were smaller and the muscle fibers were well protected after meat was treated with trehalose/COS, and thus the water was held better by the muscle fibers.

**Table 1 foods-11-01041-t001:** Definition and score criterion of sensory quality attributes.

Attributes	Definition	Score Criterion (1~9 Scale)
Color	Appearance of color and brightness of meat surface.	1 = Completely discolored, mold. 9 = Same red and bright as fresh meat.
Smell	The freshness and purity of smell from meat sample.	1 = Strong unpleasant odor. 9 = Fresh and pleasant odor.
Texture	The extent of meat sample recovering after being gently pressed by the finger.	1 = Not recover at all. 9 = Completely recover.
Overall acceptance	Rank of the samples regarding preference of overall impression.	1 = Dislike extremely. 9 = Like extremely.

**Table 2 foods-11-01041-t002:** The fitting results of various kinds of hydrogen bonds.

	Hydrogen Bond Type		Abbreviations	Wave Number/cm^−1^	Average Peak Area	Standard Deviations/%	Relative Strength/%
MP	Free hydroxyl	I	-OH	3595	70.10	5.65	4.39
	Intramolecular hydrogen bond	III	OH...OH	3409	351.85	5.19	22.04
		V	Annular polymer	3219	286.07	10.67	17.92
	Intermolecular hydrogen bond	II	OH...π	3522	216.82	4.23	13.58
		IV	OH...ether O	3295	485.58	9.27	30.42
		VI	OH...N	3132	185.66	4.23	11.63

MP + trehalose	Free hydroxyl	I	-OH	3595	29.12	6.18	1.77
	Intramolecular hydrogen bond	III	OH...OH	3409	350.78	8.67	21.29
		V	Annular polymer	3219	397.56	5.70	24.13
	Intermolecular hydrogen bond	II	OH...π	3522	148.34	8.67	9.00
		IV	OH...ether O	3295	587.25	10.48	35.64
		VI	OH...N	3132	134.73	4.16	8.18

MP + COS	Free hydroxyl	I	-OH	3595	12.75	1.53	1.37
	Intramolecular hydrogen bond	III	OH...OH	3409	218.42	6.29	23.51
		V	Annular polymer	3219	205.16	3.81	22.08
	Intermolecular hydrogen bond	II	OH...π	3522	95.85	3.79	10.31
		IV	OH...ether O	3295	321.52	5.20	34.60
		VI	OH...N	3132	75.53	3.06	8.13

**Table 3 foods-11-01041-t003:** The pH value and surface color of meat with different treatments.

	pH	Surface Color		
		*L**	*a**	*b**
meat	5.93 ± 0.03 ^a^	51.09 ± 0.09 ^a^	23.43 ± 0.14 ^a^	40.77 ± 0.29 ^a^
meat + trehalose	5.88 ± 0.03 ^b^	57.54 ± 0.08 ^b^	23.89 ± 0.34 ^a^	46.28 ± 0.36 ^b^
meat + COS	5.85 ± 0.04 ^b^	57.48 ± 0.02 ^b^	19.78 ± 0.05 ^b^	48.17 ± 0.08 ^c^

The results are mean ± standard deviation, and the letters (a–c) in the same column indicate significant differences (*p* < 0.05) between the data of different samples.

**Table 4 foods-11-01041-t004:** Sensory evaluation of meat.

	Index			
	Color	Smell	Texture	Overall Acceptance
meat	5.33 ± 0.94 ^b^	7.67 ± 0.94 ^ab^	7.00 ± 1.41 ^ab^	5.00 ± 0.82 ^bc^
meat + trehalose	5.33 ± 0.94 ^b^	8.50 ± 0.50 ^a^	8.75 ± 0.43 ^a^	7.00 ± 0.00 ^a^
meat + COS	6.25 ± 1.25 ^ab^	5.50 ± 0.50 ^c^	3.67 ± 0.47 ^c^	4.00 ± 0.82 ^c^

The results are mean ± standard deviation, and the letters (a–c) in the same column indicate significant differences (*p* < 0.05) between the data of different samples.

## Data Availability

The data presented in this study are available on request from the corresponding author.

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
