# Peer review of "Myofibrillar Protein Interacting with Trehalose Elevated the Quality of Frozen Meat"

_foods, 2022, doi:10.3390/foods11071041_

Round 1

Reviewer 1 Report

The following points should be answered. 
Line 61-63: These sentences should be combined.
Line 49: There is this sign (-) between some words. It should be removed. For example line 49, line 56, 68, etc.
Line 76: …. femorisl ???
LÄ°ne 140-142: not italic
Line 180-184: The study does not have an experimental plan.  How many times was it repeated? Completely randomized design or randomized blocks design???

Author Response

Dear reviewer,

Thank you for your valuable comments. We would like to answer these questions as follows.

1.Line 61-63:

The sentences had be combined through several phrase to make the expression continuous. The sentences after revision was presented as following.

"Chitooligosaccharide (COS) is alkaline amino oligosaccharide consisting of N-acetylglucosamine connected by β-1,4 glycosidic bonds [20]. Similar to trehalose, COS also showed well improving effect on cell resistance towards stress conditions such as chilling and salt [21-22]. In addition to the cell protective effect mentioned above, COS showed well protective effect on biomacromolecule such as proteins under stress conditions as well[20]."

2.Line 49:

The sign(-) had been removed, and the manuscript was also checked.

3.Line 76:

Typo, corrected as "biceps femoris". The samples were took from the hind leg of pork post slaughter within 48 h. This part of pig carcass was chose as experimental object because it generally storage under frozen condition.

4.Line 140-142:

The font format was set to not italic.

5.Line 180-184:

The experiment repeated 3 times, using a completely randomized design. 

To clarify the process of sensory evaluation experiment. Section 2.12 was revisioned and presented as follows.

“2.12. Sensory evaluation

Meat samples were assessed by sensory evaluation panel. The evaluating experiment was carried out in individual booths at the Sensory Analysis Laboratory, with the participation of 60 untrained judges of both male and female gender, belonging to the Meat Process& Health Innovation Team of the Hefei University of Technology. Meat samples were thawing and then immediately presented to panellists in random order. Each judge received an evaluation score form along with triplicate samples from all types of treatment, placed in a stainless steel plate. Judges were instructed to evaluate the color, smell, texture and overall acceptance on a 9-point scale. The definition and score criterion of sensory quality attributes was presented in Table 1.[31-32]. The experiment was carried out for three times.”

Thank you again for your valuable review work. Your comments will be of great help in our future research work. With all the revisions mentioned above, there still might be questions remain to be answered. We look forward to your reply.

Reviewer 2 Report

  1. English needs significant improvement
  2. Throught the manuscript some words are separated by dashes in the middle (for instance lines: 16, 17, 1, 21, 22, 24, etc.)
  3. Lines 21-22 sentence should be reformulated
  4. Lines 77, 85 biceps femorisl - typo. Please correct that.
  5. Line 101 space in the word "trehalose"
  6. Section 2.7. How many samples were in each group?
  7. Line 164. d/o light source? What was the measuring head diameter.
  8. Please add to the section 2.10 surface color. How many replicates were made?
  9. Section 2.12. what kind of questionnary was used for testing?
  10. Section 2.13. Experiment was carried once or more times? Please specify.
  11. Lack of unit in Fig. 3.
  12. Line 295 "the" repeated twice
  13. Line 361"the" repeated twice
  14. Line 408 "the" repeated twice
  15. Change references to numbers in squere brackets
  16. References sholud be formatted in accordance to the instructions provided by the Journal.

Author Response

Dear reviewer,

Thank you for your valuable comments. We would like answer these questions as follows.

1.For the improvement of English, the manuscript was submitted to editing services listed at https://www.mdpi.com/authors/english after it was revised by authors to answer the question pointed out by reviewer.

2.The sign(-) at this place had been removed, and the manuscript was also checked.

3.The sentence at line21-22 had been reformulated. The sentences after revision was presented as follows.

“Subsequently, the evaluation of effect of trehalose and COS on the physicochemical properties of frozen meat was conducted.”

4.Typo "biceps femorisl" at line77 and line85 was corrected as "biceps femoris".

5.The space in the word "trehalose" was removed.

6.Three replicates of sample were made for each type of treatment. Section 2.7 had been revised for clarify and presented as follows.

“2.7. Treatment of meat samples

After shaping and weighing, meat pieces were randomly assigned to 3 different types of treatment as follows: add trehalose (0.5 %, m/m), add COS (0.5 %, m/m), and blank control (no adding). The treating process was then carried out. The trehalose/COS were evenly sprinkle on the surface of the cuboid meat pieces. Subsequently, the meat pieces were wrapped with polyethylene (PE) film (O2 transmittance 600 cm3·m-2·d-1, moisture transmittance 5 ~ 10 g·m-2·d-1). Blank control samples were also wrapped with same PE film. The blank control, meat treated with trehalose, and meat treated with COS was tagged as meat, meat + trehalose, and meat + COS, respectively. Three replicates of sample were made for each type of treatment.

All the samples were frozen in a refrigerator (Haier, Qingdao, China) at -30 ℃ for 24 h, and then taken out of the refrigerator and thawing at 0 ~ 4 ℃ for 12 h, and the core temperature of meat raised to 0 ~ 4 ℃.”

7.The d/0 is the lighting conditions of colorimeter.The light source of colorimeterr is d/65, and measuring head diameter is ∅20mm.

8.Three replicates of sample were made. Section 2.10 had been revised for clarify and presented as follows.

“2.10. Surface color

The surface color of meat were measured according to the CIE L*a*b* color system (L* for brightness, a* for redness, and b* for yellowness) using colorimeter (WSF colorimeter, INESA Optical, Shanghai, China ) equipped with d/65 light source. Three replicates of sample were made for each type of treatment. The experiment was carried out for three times.”

9.The sensory evaluation panellists were instructed to evaluate the qualities of meat sample including color, smell, texture and overall acceptance in a 9-point scale. The stander of judgement was presented in Table 1. For example, if the color of meat sample is red and bright like fresh meat, then judges might score it as 7, 8, or even 9, basing on the score instruction form and their judgement. 

Section 2.12 had been revised for clarify and presented as follows.

“2.12. Sensory evaluation

Meat samples were assessed by sensory evaluation panel. The evaluating experiment was carried out in individual booths at the Sensory Analysis Laboratory, with the participation of 60 untrained judges of both male and female gender, belonging to the Meat Process& Health Innovation Team of the Hefei University of Technology. Meat samples were thawing and then immediately presented to panellists in random order. Each judge received an evaluation score form along with triplicate samples from all types of treatment, placed in a stainless steel plate. Judges were instructed to evaluate the color, smell, texture and overall acceptance on a 9-point scale. The definition and score criterion of sensory quality attributes was presented in Table 1.[31-32]. The experiment was carried out for three times.”

10.Section 2.13. Experiment was carried 3 times. Section 2.13 had been revised for clarify and presented as follows.

“2.13. Statistical analysis

Statistical analysis of results was performed using SPSS software (SPSS Inc., Ver.26). All data was presented as means ± standard deviation (SD). One-way analysis of variance (ANOVA) was employed to determine the statistical difference. Significant differences between means were identified using Duncan’s multiple range test (P < 0.05). At least three replicates of sample were made for each type of treatment in experiment. The experiment was carried out for three times.”

11.Fig. 3 after being revised was presented as following.

12.The sentence at line 295 was revised and presented as follows.

“The introduction of trehalose/COS reduced the surface zeta potential of MP, thereby reduced the electrostatic repulsion of MP.”

13.The sentence at line 361 was revised and presented as follows.

This may be due to that trehalose/COS made the juice ooze out of meat surface, which enhanced the reflection of light (Faieta et al., 2020).

14.The sentence at line 408 was revised and presented as follows.

In Fig. 7 (a), the muscle fibers in natural state without addition of trehalose or COS were arranged tightly and orderly well in parallel to each other.

15.The references had been changed. A sample of the manuscript after revision was presented.

“Because it can inhibit the proliferation of spoilage microorganisms [1-2].”

16.References had been formatted. A sample of the manuscript was being revision was presented.

  1. “Jiang,J.; Tang, X.Y.; Xue, Y.; Lin, G.; Xiong, Y.L.L. Dietary linseed oil supplemented with organic selenium improved the fatty acid nutritional profile, muscular selenium deposition, water retention, and tenderness of fresh pork. Meat Sci. 2017, 131, 99-106; DOI: 10.1016/j.meatsci.2017.03.014.”

Thank you again for your valuable review work. Your comments will be of great help in our future research work. With all the revisions mentioned above, there still might be questions remain to be answered. We look forward to your reply.

Reviewer 3 Report

The manuscript describes a study on improving frozen meat quality parameters with the application of two compounds (disaccharide and oligosaccharide). The topic is relevant. The paper is well structured, and the scientific results are adequate. The methodology is well specified and delineated. The analysis methods are coherent with the proposed objectives.

Extensive editing of English language and style is required (Line 31: “microorgan-isms”; Line 73: “evalu-ated”). The reading of the text was not fluid, and the authors can improve it. For example, the authors present figure 1 several times (lines 189 197, 204).

Author Response

Dear reviewer,

Thank you for your valuable comments. We would like answer these questions as follows.

1.For the improvement of English, the manuscript was submitted to editing services listed at https://www.mdpi.com/authors/english after it was revised by authors to answer the question pointed out by reviewer. Several sentences in text had been revised by author for clarify and better expression. Section 2.7 was presented for example as follows.

“2.7. Treatment of meat samples

After shaping and weighing, meat pieces were randomly assigned to 3 different types of treatment as follows: add trehalose (0.5 %, m/m), add COS (0.5 %, m/m), and blank control (no adding). The treating process was then carried out. The trehalose/COS were evenly sprinkle on the surface of the cuboid meat pieces. Subsequently, the meat pieces were wrapped with polyethylene (PE) film (O2 transmittance 600 cm3·m-2·d-1, moisture transmittance 5 ~ 10 g·m-2·d-1). Blank control samples were also wrapped with same PE film. The blank control, meat treated with trehalose, and meat treated with COS was tagged as meat, meat + trehalose, and meat + COS, respectively. Three replicates of sample were made for each type of treatment.

All the samples were frozen in a refrigerator (Haier, Qingdao, China) at -30 ℃ for 24 h, and then taken out of the refrigerator and thawing at 0 ~ 4 ℃ for 12 h, and the core temperature of meat raised to 0 ~ 4 ℃.”

2.The sign(-) at line 31, line 73 had been removed, and the manuscript was also checked.

3.The way that Fig.1being presented was changed. Line 189, line 197, and line 204 were revised and presented as follows.

“The Fourier-transformed infrared spectra (FTIR) of the trehalose, COS, MP, MP + trehalose, and MP+COS were showed in Fig. 1.”

“For MP, its characteristic absorption peaks ...”

“For MP + trehalose and MP + COS, the FTIR spectrum of trehalose and COS has been removed by”

Thank you again for your valuable review work. Your comments will be of great help in our future research work. With all the revisions mentioned above, there still might be questions remain to be answered. We look forward to your reply.